# Parametric model reduction of mean-field and stochastic systems via higher-order action matching

**Jules Berman**[*]
Courant Institute of
Mathematical Sciences
New York University
New York, NY 10012
`jmb1174@nyu.edu`

**Tobias Blickhan**[*]
Courant Institute of
Mathematical Sciences
New York University
New York, NY 10012
`tobias.blickhan@nyu.edu`

**Benjamin Peherstorfer**
Courant Institute of Mathematical Sciences
New York University
New York, NY 10012
`pehersto@cims.nyu.edu`

## Abstract

The aim of this work is to learn models of population dynamics of physical systems that feature stochastic and mean-field effects and that depend on physics parameters. The learned models can act as surrogates of classical numerical models to efficiently predict the system behavior over the physics parameters. Building on the Benamou-Brenier formula from optimal transport and action matching, we use a variational problem to infer parameter- and time-dependent gradient fields that represent approximations of the population dynamics. The inferred gradient fields can then be used to rapidly generate sample trajectories that mimic the dynamics of the physical system on a population level over varying physics parameters. We show that combining Monte Carlo sampling with higher-order quadrature rules is critical for accurately estimating the training objective from sample data and for stabilizing the training process. We demonstrate on Vlasov-Poisson instabilities as well as on high-dimensional particle and chaotic systems that our approach accurately predicts population dynamics over a wide range of parameters and outperforms state-of-the-art diffusion-based and flow-based modeling that simply condition on time and physics parameters.

## 1 Introduction

Predicting the behavior of time-dependent processes $X_{t,\mu}$ over time $t$ and across varying physics parameters $\mu$ is a key challenge in computational science and engineering [46, 65]. The dynamics of $X_{t,\mu}$ typically are described by systems of (stochastic) differential equations, which are derived from physics models and can be computationally expensive to simulate [40, 32]. Thus, it is desirable to learn reduced or surrogate models that can be rapidly evaluated to predict the system behavior across varying physics parameters [72, 10, 11, 45].

---

[*]Equal contribution

38th Conference on Neural Information Processing Systems (NeurIPS 2024).

**Reduced modeling via learning population dynamics**   Given a data set of samples, i.e., realizations of the random variable $X_{t,\mu}$ on a suitable domain $\mathcal{X} \subseteq \mathbb{R}^d$,

$$\{X^i_{t_j,\mu_k} \,|\, i = 1, \ldots, N_x, \quad j = 1, \ldots, N_t, \quad k = 1, \ldots, N_\mu\} \subset \mathcal{X}, \tag{1}$$

we aim to learn a dynamical-system reduced model to rapidly predict samples that approximately follow the same law $\rho_{t,\mu}$ as $X_{t,\mu}$ over time $t$ and varying physics parameter $\mu$. We refer to the evolution of $\rho_{t,\mu}$ in time as population dynamics. Learning the population dynamics instead of learning the dynamics of the individual trajectories $t \mapsto X^i_{t,\mu}$ for all $i = 1, \ldots, N_x$ and $\mu$ can be beneficial: There are cases where $\rho_{t,\mu}$ does not change in time, yet every sample trajectory $t \mapsto X^i_{t,\mu}$ follows complicated dynamics. For example, consider incompressible fluid dynamics with constant density. Samples corresponding to particles that comprise the fluid can have complicated trajectories, whereas on a distribution level, the density of the fluid is constant and so are the population dynamics. Furthermore, learning population dynamics seamlessly treats deterministic and stochastic systems because the stochastic models that we consider can be expressed as deterministic Fokker-Planck equations on the population level.

**Our approach: Learning parametric minimal energy vector fields that represent population dynamics**   Building on standard literature on optimal transport theory [8] as well as the so-called action-matching loss introduced in [61], we pose a variational problem to learn gradient fields $\nabla s_{t,\mu}$ so that the continuity equation corresponding to the vector field given by $\nabla s_{t,\mu}$ approximates the population dynamics $\rho_{t,\mu}$ of the samples (1). In the spirit of reduced modeling [72, 10, 11, 45], we seek a vector field $s_{t,\mu}$ that generalizes to different values of the physics parameters $\mu$. We therefore optimize for $s_{t,\mu}$ that minimizes the average objective of a variational problem over all parameters $\mu \sim \nu$, where $\nu$ describes the distribution of parameters on the domain $\mathcal{D} \subset \mathbb{R}^p$. We parametrize $s_{t,\mu}$ with a neural network with weight modulation [39, 13] so that it can be evaluated quickly over $t$ and $\mu$.

*Rapid sample generation in inference phase* Predictions at inference time at new physics parameters $\mu$ are made by sampling based on the vector field $\nabla s_{t,\mu}$, which means that our approach represents $\rho_{t,\mu}$ through the application of $\nabla s_{t,\mu}$ on an initial condition. Importantly, time $t$ in the inference step corresponds to the time of the physics problem so that in one inference step a whole sample trajectory is obtained, rather than a sample at one specific time point as in regular conditioning-based methods (see literature review). Thus, we can rapidly generate samples that follow the law $\rho_{t,\mu}$ in the inference phase.

*Stabilizing training with higher-order quadrature* An important part of our contribution is stabilizing the training procedure by accurately estimating the objective of the variational problems from few data samples. In particular, instead of uniformly sampling over the data (1), we introduce an empirical loss (8) that utilizes higher-order quadrature [27] in the time direction so that the learned $\nabla s_{t,\mu}$ accurately captures the dynamics over time $t$. Consequently, we refer to our approach as higher-order action matching (HOAM). Our numerical experiments show that the higher-order quadrature in the empirical loss is key for learning gradient fields $\nabla s_{t,\mu}$ that accurately capture the evolution in time $t$ and that generalize across physics parameters $\mu$.

**Literature review**   We review relevant literature; see Figure 1 for an overview.

*Non-intrusive and data-driven surrogate modeling* There is a range of surrogate and latent modeling methods that aim to learn or reduce the sample dynamics of the realizations rather than the population dynamics, such as dynamic mode decomposition, Koopman-based methods, and others [71, 76, 86, 12, 46, 58, 92] as well as neural network-based methods such as neural ordinary differential equations [19, 28, 48]. There also are methods for stochastic systems [51, 42, 88, 19, 28, 73, 21]. However, all of these methods ignore physics parameter dependencies and/or aim to learn the sample dynamics, whereas we focus on parametric population dynamics.

*Population dynamics and trajectory inference* Learning population dynamics has been considered extensively in computational biology in the context of gene expression, where the focus is on learning from independent samples at selected time points rather than from sample trajectories [34, 30, 93, 75, 85, 47]; however, many of these approaches [17, 84] are

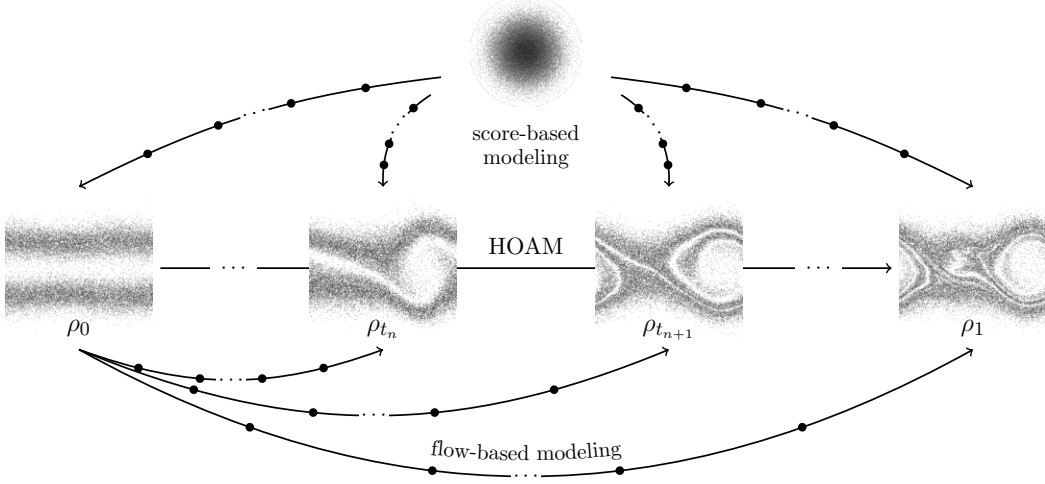

Figure 1: Parametric model reduction with our HOAM seeks to learn vector fields that represent population dynamics $\rho_t$ over time $t$. In contrast, parametric model reduction with score-based diffusion denoising and flow-based modeling requires conditioning on time $t$, which leads to separate, costly inference steps for each time step of a sample trajectory.

simulation-based and thus require integrating dynamics during the training or parameterizing the density additionally to the vector field. These works also are not concerned with generalizing over a range of physics parameters in many cases.

*Diffusion- and flow-based modeling* There is a large body of work on diffusion-based [91, 79, 36, 41, 81, 82] and flow-based modeling [2, 54]; see [1] for a detailed review. These approaches are not taking into account time $t$ because they learn paths between a reference and a target distribution only. There are works that condition on time $t$ and a parameter $\mu$ such as [68, 14, 26, 37, 33, 38, 52], but this requires then generating a path for each time step at inference time, which is computationally expensive. Furthermore, the conditioning on time $t$ means that the target distribution $\rho_{t,\mu}$ at each time $t$ and $\mu$ is different, and thus a separate hyper-parameter tuning can be required, which is impractical over many time steps and physics parameters as in our physics problems; see our numerical experiments. The works [15, 78, 50] compute transport-based solutions but parametrize different quantities than our approach, require actively sampling data, and ignore physics parameters $\mu$. We note that there also is work on forecasting with diffusion- and flow-based modeling [68, 62, 18, 20], which is a different task than our task of predicting across varying physics parameters.

*Optimal transport* Besides the machine learning literature, variational approaches for inferring vector fields are extensively used in optimal transport theory [5, 4]. Of particular importance to us is the formulation by Benamou and Brenier [8]. The Bennamou-Brenier formula describes a joint optimization problem over vector fields and paths in probability space and the action matching loss [61] is the restriction of this optimization problem to the case of a fixed path and the vector field parametrized by a neural network, which are core building blocks for us that we show can be used together with a parameter dependency.

**Contributions**  We summarize our contributions:

**(a)** Developing a loss to learn population dynamics that remain valid across varying physics parameters by building on optimal transport literature [8] and action matching [61].
**(b)** Introducing higher-order quadrature schemes for the loss to efficiently couple the gradient fields over time. This leads to lower variance estimators of the loss that critically stabilize training.
**(c)** Demonstrating on a range of physics problems from Vlasov-Poisson instabilities to high-dimensional chaotic systems that our approach leads to (i) accurate predictions of population dynamics and (ii) orders of magnitude speedups in inference/prediction over

classical methods that numerically solve the underlying partial differential equations as well as standard diffusion- and flow-based models that condition on physical time.

We provide an implementation of our method at `https://github.com/julesberman/HOAM`.

## 2 Method

### 2.1 Parameter-dependent population dynamics

**Continuity equation** Let us consider data (1) corresponding to the probability measure $\rho_{t,\mu}$, which is absolutely continuous for $t \in [0,1]$ and $\mu \in \mathcal{D}$. We use the same notation for the measure and its density. The density $\nu$ of $\mu$ is also assumed to be absolutely continuous on $\mathcal{D}$. We consider population dynamics of $X_{t,\mu} \sim \rho_{t,\mu}$ that can be described by the continuity equation

$$\partial_t \rho_{t,\mu} = -\nabla \cdot (\rho_{t,\mu} v_{t,\mu}), \qquad \text{for all } t \in [0,1], \mu \in \mathcal{D}, \tag{2}$$

with the initial condition $\rho_{t=0,\mu} =: \rho_{0,\mu}$ and vector field $v_{t,\mu}$. Notice that in our case the continuity equation (2) depends on the physics parameter $\mu \sim \nu$. There can be many vector fields $v_{t,\mu}$ that lead to the same population dynamics (2). For example, if $v_{t,\mu}$ is a vector field that describes the dynamics of $\rho_{t,\mu}$ via (2), then another vector field is given by $v'_{t,\mu} = v_{t,\mu} + w/\rho_{t,\mu}$ with any other $w$ that satisfies $\nabla \cdot w = 0$ as long as $\rho_{t,\mu}$ is positive.

**Uniqueness via gradient fields and the corresponding elliptic problems** Because we aim to learn a vector field from sample data (1) that describes the population dynamics (2) of the corresponding law $\rho_{t,\mu}$, it is helpful to remove this non-uniqueness. One way to do so is to restrict the vector field to $v_{t,\mu} = \nabla s_{t,\mu}$ so that it is a gradient field [4, p. 45]. Plugging $v_{t,\mu} = \nabla s_{t,\mu}$ into (2), together with the assumptions $\rho_{t,\mu} > 0$ and $\int_{\mathcal{X}} \partial_t \rho_{t,\mu} \mathrm{d}x = 0$, leads to parametric elliptic problems in $s_{t,\mu}$

$$-\nabla \cdot (\rho_{t,\mu} \nabla s_{t,\mu}) = \partial_t \rho_{t,\mu}, \tag{3}$$

with coefficient function $\rho_{t,\mu}$, right-hand side (source term) $\partial_t \rho_{t,\mu}$, and homogeneous Neumann boundary conditions $\rho_{t,\mu} \nabla s_{t,\mu} \cdot \hat{n} = 0$ on $\partial \mathcal{X}$ with normal vector $\hat{n}$ for all $t \in [0,1]$ and $\mu \in \mathcal{D}$. The weak forms of the elliptic problems (3) lead to energy minimization problems that can be used to learn the gradient field $s_{t,\mu}$ via optimization:

$$\min_{s \in H^1(\rho_{t,\mu}, \mathcal{X})} E_{t,\mu}(s) := \min_{s \in H^1(\rho_{t,\mu}, \mathcal{X})} \frac{1}{2} \int_{\mathcal{X}} |\nabla s|^2 \rho_{t,\mu} \mathrm{d}x - \int_{\mathcal{X}} \partial_t \rho_{t,\mu} s \mathrm{d}x \tag{4}$$

for each $t \in [0,1]$ and $\mu \in \mathcal{D}$. The space $H^1(\rho_{t,\mu}, \mathcal{X})$ contains functions $s$ with $\int_{\mathcal{X}} |\nabla s|^2 \rho_{t,\mu} \mathrm{d}x < \infty$, which is the energy (semi-)norm corresponding to the $\rho_{t,\mu}$-weighted inner product [29, Sec. 2.3.2].

**Optimal transport** Standard elliptic theory guarantees unique solutions up to constants of (4) in the Sobolev space $H^1(\mathcal{X})$ under strong assumptions on $\rho_{t,\mu}$ such as uniform boundedness by a positive constant for all $t$ and $\mu$; see [29, Proposition 2.2] and [11, Section 3.2]. The theory of optimal transport allows treating the much more general case when $\rho_{t,\mu}$ is not uniformly bounded away from zero and possibly atomic; we refer to [8] and [74, Section 5.3.1] for details. Among all vector fields $v_{t,\mu}$ that are compatible to $\rho_{t,\mu}$ in the sense of (2), gradient fields $\nabla s_{t,\mu}$ have the smallest associated kinetic energy $\frac{1}{2} \int_{\mathcal{X}} |v|^2 \rho_{t,\mu} \mathrm{d}x$, which is the objective considered in [8]. In the language of optimal transport and in particular the formalism of [63], vector fields with minimal kinetic energy describe tangent vectors to the curve $t \mapsto \rho_{t,\mu}$. The metric is the inner product of $L^2(\rho_{t,\mu}, \mathcal{X}, \mathbb{R}^d)$. This is the weak Riemannian structure of $\mathcal{P}(\mathcal{X})$ equipped with the Kantorovich-Rubinstein metric and described in detail in [5, Chapter 8]. We give a short description in Appendix E.

**Energy functional with entropy term** Instead of the energy (4), we can also use other choices of the energy to select gradient fields, as long as energy functions are convex to maintain uniqueness. We consider an energy that is based on a different notion of discrepancy on $\mathcal{P}(\mathcal{X})$, the entropic optimal transport or Schrödinger bridge problem [77, 56],

$$E_{t,\mu}^{\epsilon}(s) = \frac{1}{2} \int_{\mathcal{X}} |\nabla (s - \frac{\epsilon^2}{2} \log \rho_{t,\mu})|^2 \rho_{t,\mu} \mathrm{d}x - \int_{\mathcal{X}} \partial_t \rho_{t,\mu} s \mathrm{d}x, \tag{5}$$

which depends on $\epsilon \geq 0$. The energy $E_{t,\mu}^{\epsilon}$ is of particular interest for two reasons: One, the Euler-Lagrange equation of (5) in strong form is the Fokker-Planck equation for $s_{t,\mu}^{\epsilon}$: $\partial_t \rho_{t,\mu} = -\nabla \cdot (\rho_{t,\mu} \nabla s_{t,\mu}^{\epsilon}) + \frac{\epsilon^2}{2} \Delta \rho_{t,\mu}$, again with homogeneous Neumann boundary conditions for all $t \in [0,1]$ and $\mu \in \mathcal{D}$; see Appendix C. This means we can efficiently generate samples after learning $s_{t,\mu}^{\epsilon}$ via corresponding stochastic differential equations (SDEs). Two, it can be interpreted as regularizing the field $s_{t,\mu}$, which we discuss in Appendix C.

## 2.2 Loss for learning vector fields over time $t$ and physics parameter $\mu$

**Variational formulation over $t$ and $\mu$**  So far we just carried along time $t$ and physics parameter $\mu$ but did not address them in the variational problems, i.e., we had separate variational problems (4) for all $t \in [0,1]$ and $\mu \sim \nu$. We now propose to consider the average energy over $t$ and $\mu$ to infer a map $s : [0,1] \times \mathcal{D} \to H^1(\rho_{t,\mu}, \mathcal{X}), (t,\mu) \mapsto s_{t,\mu}$, which is called a solution map in reduced modeling [72, 10, 11, 45],

$$\min_{s:[0,1]\times\mathcal{D}\to H^1(\rho_{t,\mu},\mathcal{X})} E^{\epsilon}(s) := \min_s \int_{\mathcal{D}} \int_0^1 E_{t,\mu}^{\epsilon}(s_{t,\mu}) \, \mathrm{d}t \, \mathrm{d}\nu(\mu). \tag{6}$$

Notice that time $t$ and physics parameter $\mu$ have two different effects on the gradient field $\nabla s_{t,\mu}$: Time $t$ couples the elliptic problems (i.e., (3) for $\epsilon = 0$) via the time derivative $\partial_t \rho_{t,\mu}$; see Appendix D. In contrast, the elliptic problems are uncoupled over $\mu$ and can be considered separately. This means that to compute the solution to an elliptic problem for one value of $\mu \in \mathcal{D}$, one does not need to consider any other $\mu' \in \mathcal{D}$. This will allow us to sample the physics parameters over $\mathcal{D}$ independently from each other when estimating the corresponding loss, whereas we will use higher-order quadrature to obtain an accurate approximation of the time integral to ensure the coupling between the time points is reflected in $s_{t,\mu}$; see Section 2.3.

**Loss for learning gradient fields from samples over $t$ and $\mu$**  The energy $E_{t,\mu}$ defined in (4) as well as the energy $E_{t,\mu}^{\epsilon}$ defined in (5) leads to a loss that can be estimated from samples (1). The quantity $\partial_t \rho_{t,\mu}$ appears in (4) and (5), which is typically unavailable when we have access to data samples (1) only. Integration by parts of the term involving $\partial_t \rho_{t,\mu}$ eliminates it, see also Appendix D. We arrive at

$$E^{\epsilon}(s) = \int_{\mathcal{D}} \left[ \int_0^1 \int_{\mathcal{X}} \left( \frac{1}{2}|\nabla s_{t,\mu}|^2 + \partial_t s_{t,\mu} + \frac{\epsilon^2}{2}\Delta s_{t,\mu} \right) \rho_{t,\mu} \mathrm{d}x \mathrm{d}t - \int_{\mathcal{X}} s_{t,\mu}\rho_{t,\mu}\mathrm{d}x \Big|_{t=0}^{t=1} \right] \mathrm{d}\nu(\mu). \tag{7}$$

Note that this loss is comprised only of expectation values with respect to $\rho_{t,\mu}$ and is therefore well-defined also for empirical distributions. The choice $\varepsilon > 0$ assumes that the Fisher information of $\rho_{t,\mu}$ is finite.

**Remark 1.** *Loss functions of the form as (7) but without the parameter dependence have been used in [61] and [47, Theorem 2.1]. In fact, the case with $\epsilon = 0$ appears already in [8, Equation 35] and [64, Section 3]. We build on these results but work with population dynamics that depend on physics parameters, which leads to the loss shown in (7).*

## 2.3 Parameterizing the vector field, estimating the loss from data, sampling

**Parametrizing $s_{t,\mu}$ with weight modulations**  We parametrize the vector field $s_{t,\mu}$ via a neural network with continuous versions of low-rank adaptation (CoLoRA) layers, which have been successfully used for parametric model reduction of deterministic time-dependent dynamical systems [13]; see also [39]. The layers have the form $\mathcal{C}(x) = Wx + \phi(t,\mu)ABx + b$, where $W$ is a weight matrix, $A, B$ are low-rank matrices, $b$ is a bias vector, and $\phi(t,\mu) \in \mathbb{R}$ is a scalar weight modulation; see Appendix B. Only the weight modulations $\phi(t,\mu)$ depend on time $t$ and physics parameter $\mu$. We use a hyper-network $h : [0,1] \times \mathcal{D} \times \Psi \to \mathbb{R}$ that depends on the weight vector $\psi \in \Psi \subseteq \mathbb{R}^q$ to map $t$ and $\mu$ to the modulation weights $\phi(t,\mu) = h(t,\mu;\psi)$. The weights $W, A, B, b$, which are independent of $t$ and $\mu$, over all layers are collected into the weight vector $\theta \in \Theta \subseteq \mathbb{R}^{q'}$. Typically $q \ll q'$. Using the hyper-network encourages continuity of $s_{t,\mu}$ in time $t$, which is key for many physics problems [13].

**Combining higher-order quadrature and Monte Carlo sampling for estimating the loss from sample data** Estimating the loss (7) from data can be challenging because the three nested integrals (expectations) over the samples $X_{t,\mu}^i$, time $t$, and physics parameter $\mu$ can have different properties and correspondingly need different numerical treatment. Our numerical results show that it is critical to accurately estimate the loss to avoid instabilities in the training; see Section 3 and Figure 2.

We propose a combination of higher-order numerical quadrature and Monte Carlo sampling to estimate the loss (7). In particular, we propose to use a higher-order quadrature rule for the time $t$ integral. Because it is a one-dimensional integral, standard higher-order quadrature rules from numerical analysis are applicable [27]. The time integral needs to be estimated with particular high accuracy to ensure the coupling between the time points as well as the coupling to the boundary terms to match the path from $\rho_{0,\mu}$ at time $t = 0$ to $\rho_{1,\mu}$ at time $t = 1$. Our numerical results will show that estimating the time integral to high accuracy is essential for stabilizing the training. In contrast to the one-dimensional integral over time, the integrals over $\mathcal{X}$ and the parameter domain $\mathcal{D}$ can be high dimensional and thus we estimate them via Monte Carlo estimation.

We consider two high-order quadrature rules, composite Simpson's quadrature and Gauss-Legendre quadrature [27]; see Appendix A. We refer to our method as HOAM-S and HOAM-G when using either quadrature, respectively. Importantly, these quadrature rules require samples on specifically spaced time points, equidistant in the case of Simpson's and at the Gauss-Legendre nodes in the case of Gauss quadrature. If the data set (1) does not contain samples at these time points then we interpolate the data to the appropriate times. We note that for Simpson's quadrature, interpolation is typically unnecessary as data simulated with numerical methods often come at equispaced points in time.

We denote a Monte Carlo estimate of an expectation value obtained from a mini-batch as $\hat{\mathbb{E}}_{x\sim\rho}^n[f] := \sum_{i=1}^n f(X^i)$ where $X^1, X^2, \ldots, X^n \sim \rho$. Then, the empirical loss with mini-batching of sizes $n_x, n_\mu$ and $n_t$ quadrature points in time is given by

$$\hat{E}^\epsilon(s) = \hat{\mathbb{E}}_{\mu\sim\nu}^{n_\mu}\left[\sum_{n=1}^{n_t} w_n \hat{\mathbb{E}}_{x\sim\rho_{t_n,\mu}}^{n_x}\left[\frac{1}{2}|\nabla s_{t_n,\mu}|^2 + \partial_t s_{t_n,\mu} + \frac{\epsilon^2}{2}\Delta s_{t_n,\mu}\right] - \hat{\mathbb{E}}_{x\sim\rho_{t,\mu}}^{n_x}[s_{t,\mu}]\Big|_{t=0}^{t=1}\right] \quad (8)$$

where $w_n$ are numerical quadrature weights and $t_n$ are the corresponding nodes; see Appendix A for the Simpson's quadrature and Gauss-Legendre weights and nodes.

**Rapid predictions (inference) with learned reduced models** Making predictions in the inference step means drawing samples that follow the law represented by the learned gradient field $\nabla s_{t,\mu}$, which approximates the law $\rho_{t,\mu}$ of $X_{t,\mu}$. Because we train with the loss (7), we integrate the SDE $d\hat{X}_{t,\mu} = \nabla s_{t,\mu}(\hat{X}_{t,\mu})dt + \epsilon dW_t$, where $W_t$ are Wiener processes and $\epsilon$ is the same $\epsilon$ that is used in the training loss (7); see Appendix C. As initial condition, we use samples from $\rho_{0,\mu}$ at time $t = 0$. Of course other sampling schemes can be used [70].

Notice that the time $t$ in the SDE used for generating samples is the same time as of the physics problem and thus of the sample trajectory. This means that the costs of the inference step of our HOAM for generating a trajectory of length $K$ scales as $\mathcal{O}(K)$. In contrast, introducing a conditioning on time and physics parameter in, e.g., noise-conditioned score matching (NCSM) [80] and conditional flow matching (CFM) or stochastic interpolants [2, 54] requires inferring a separate sampling path for each $t$ and $\mu$ pair of interest. In particular, the inference costs of CFM scale as $\mathcal{O}(K\tau)$, where $\tau$ is the number of steps taken in the differential equation for generating one sample at one time point. For NCSM with annealed Langevin sampling, the inference costs scale as $\mathcal{O}(K\tau\sigma)$, where $\sigma$ is the number of annealing steps. Contrasting this to the scaling of $\mathcal{O}(K)$ of our HOAM approach shows that HOAM is well suited for fast predictions over $t$ and $\mu$ as required in parametric model reduction.

## 3 Numerical experiments

**Examples** We consider the following parametric dynamical systems; details in Appendix B.
*1. Harmonic oscillator:* A collection of particles evolves in four-dimensional phase-space

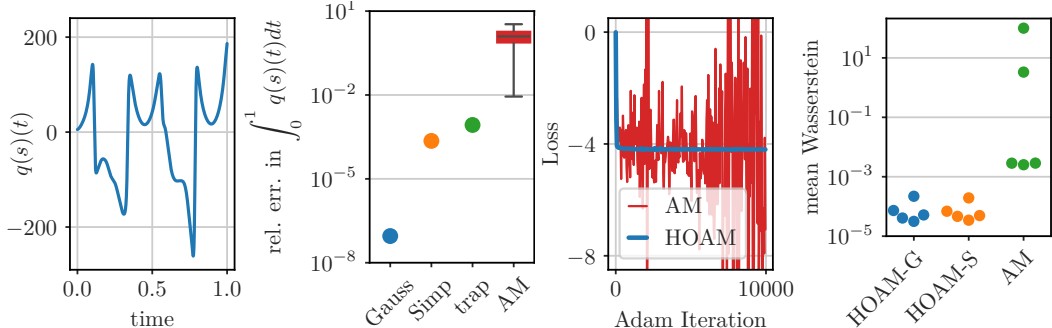

Figure 2: **Left**: During training the high-variance function $q(s)(t)$ needs to be numerically integrated for estimating the loss. **Center left**: The high variance leads to inaccurate estimates of the time integral by Monte Carlo, whereas higher-order numerical quadrature produces accurate estimates. **Center right**: Numerical quadrature in HOAM leads to stable estimates of the loss whereas Monte Carlo integration in AM leads to unstable behavior. **Right**: HOAM based on higher-order quadrature is stable and more accurate than AM.

subject to a quadratic potential $V(x) = -\frac{1}{2}\omega^2|x|^2$. In the experiments shown $\omega = 8$. The particles are initially at rest and follow a normal Gaussian distribution in space with mean $m_0 = [1, 1]$. To avoid the formation of a singularity at $\omega t = \frac{1}{2}\pi$, we add white noise of strength $\eta = 5 \times 10^{-2}$ to the momentum equation. For the case $\eta = 0$, we have analytical expressions for $\rho$ and $s$, see Appendix B.1.

*2. Two-stream instability* We numerically solve the Vlasov-Poisson partial differential equations using a particle-in-cell method to generate samples (1). We consider the two-stream instability [22, 43] in a 1D1V configuration with collisions [87, Sec 2(b)(i)], with $\beta = 10^{-3}$ and $v_0 = 1$ as in [49]. These collisions lead to stochastic sample trajectories. The parameter $\mu \in [1.2, 1.9]$ is a normalization constant related to the Debye length [83]. It controls the ratio between electric and inertial effects in the simulation.

*3. Bump-on-tail instability* Using the same numerical setup of the Vlasov-Poisson equation as for the two-stream instability, we also consider the the bump-on-tail instability [7, 35, 43]. The parameter varies as $\mu \in [1.3, 2.0]$.

*4. Strong Landau damping* We consider the strong Landau damping phenomenon that is governed by Vlasov-Poisson partial differential equations again but now in a 3D3V (six-dimensional) setup. A perturbation in the $x_1$-direction leads to the formation of phase-space structures [59]. The parameter $\mu \in [0.5, 1.5]$ is the mass of the charged particles.

*5. High-dimensional chaos* A Rayleigh–Bénard convection leads to a density gradient that sets a fluid in motion. We consider a nine-dimensional dynamical system that is derived from such a flow, which exhibits cascades that lead to chaos [69]. The parameter $\mu \in [13.7, 14.4]$ is the reduced Rayleigh number.

*6. Particles in aharmonic trap* We consider 50 particles in an aharmonic trap [16], which lead to 100-dimensional samples $X^i_{t,\mu}$ that encode the positions of the particles. The particle positions are governed by a stochastic differential equation. The parameter $\mu \in [0.3, 0.9]$ controls the velocity of the trap.

**Setup** We compare our higher-order action matching (HOAM) to the original version of action matching (AM) [61], where we handle the parameter dependence on $\mu$ in the same way as in our approach. Additionally, we compare to noise-conditioned score matching (NCSM) where samples are generated via annealed Langevin dynamics [80] and conditional flow matching (CFM) [2, 54], for which we condition on time $t$ and $\mu$; see Appendix B.

**HOAM stabilizes training with higher-order quadrature** In Figure 2 we consider the harmonic oscillator example. We learn the field $s_t$ and plot $q(s)(t) = \hat{\mathbb{E}}^{n_\mu}_{\mu\sim\nu}\hat{\mathbb{E}}^{n_x}_{x\sim\rho_{t,\mu}}[\frac{1}{2}|\nabla s_{t,\mu}|^2 +$

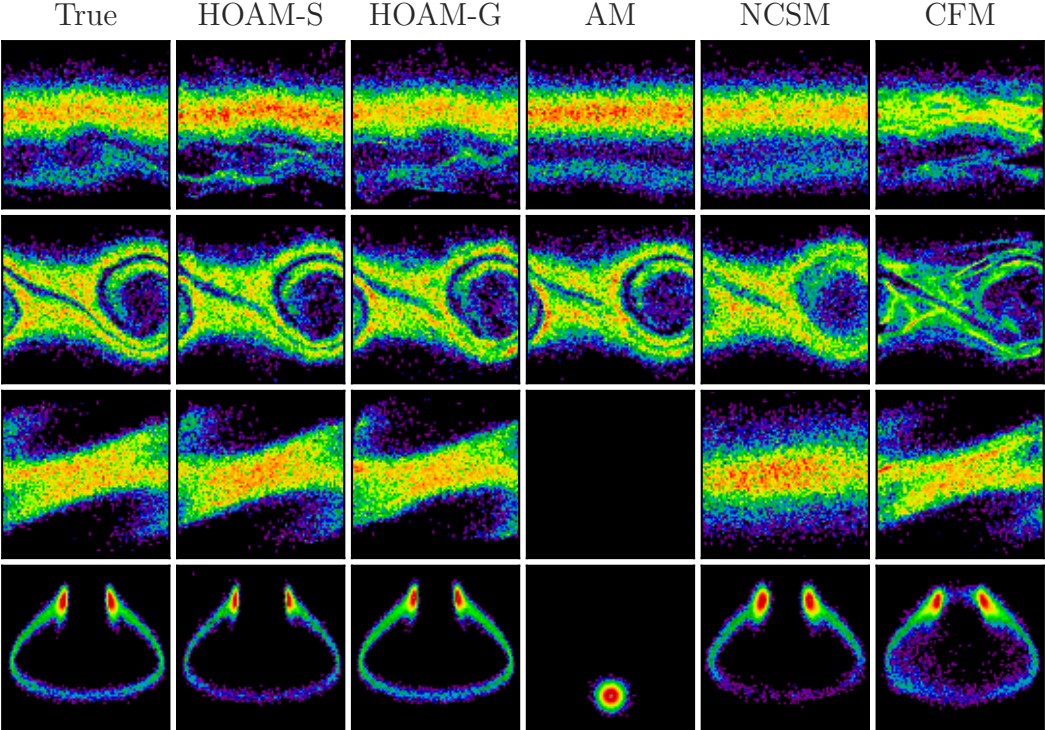

| True | HOAM-S | HOAM-G | AM | NCSM | CFM |
|------|--------|--------|-----|------|-----|

Figure 3: Histograms of solution fields. **Top**: Bump-on-tail ($t = 20$) instability. **Middle top**: two-stream ($t = 20$) instability. **Middle bottom**: Strong Landau damping ($t = 4$) instability. HOAM with Simpson's and Gauss quadrature accurately predicts the fine scale features and multi-modality of the population density in the Vlasov problems. AM does not converge on the 6 dimensional problem. **Bottom**: High-dimensional chaos [69] ($t = 3.7$, dim 3 vs dim 9). HOAM accurately predicts the low probability region that connects the two high probability regions while AM does not converge.

$\partial_t s_{t,\mu} + \frac{\epsilon^2}{2} \Delta s_{t_n,\mu}]$ over time $t$, which is the function that needs to be integrated in time to estimate the loss (7). As Figure 2 (left) shows, this function is far from smooth and exhibits several sharp peaks, which make estimating the loss challenging. In AM [61], the time integral is estimated by averaging samples uniformly taken in time, which is equivalent to Monte Carlo estimation. Figure 2 (center left) shows the relative error in estimating the time integral using Monte Carlo integration as done by AM versus numerical quadrature as in our HOAM. The trapezoidal rule, the composite Simpson's rule, and Gauss-Legendre quadrature all produce highly accurate estimates, whereas Monte Carlo integration yields inaccurate estimates of the integral with high variance.

Poor numerical estimates by Monte Carlo lead to unstable and inaccurate estimates of the loss function in AM, which eventually causes the optimization to diverge as shown in Figure 2 (center right). In contrast, our HOAM, where training is done with higher-order quadrature (in this case Simpsons' rule), the loss curve is stable and of low variance. In Figure 2 (right) we plot the mean Wasserstein distance over time of solutions generated via a gradient field $s$ trained with with Simpson's (HOAM-S), Gauss quadrature (HOAM-G), and with Monte Carlo (AM) for five seeds, which determine the random initialization of the neural network. For some seeds, AM yields reasonable solutions while for others numerically instabilities lead AM to fail. In contrast, our quadrature-based HOAM is consistently stable and provides orders of magnitude more accurate results.

**Accurate predictions with speedups for Vlasov-Poisson equations**   Our Vlasov-Poisson problems describe the interaction of charged particles with dynamics that depend on all other particles, which leads to mean-field dynamics for large numbers of particles $N_x$. Thus, reduced modeling with HOAM is well suited for this problem because the natural

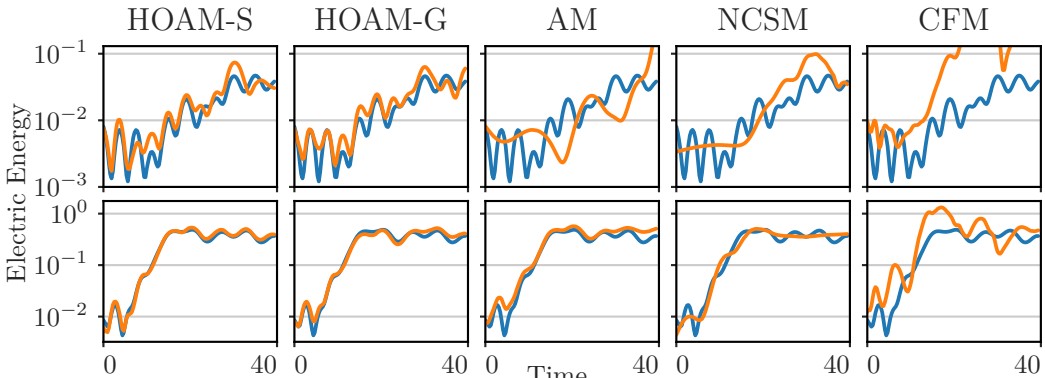

Figure 4: Electric energy of bump-on-tail (top) and two-stream (bottom) instability. HOAM with Simpson's and Gauss quadrature accurately predicts the energy growth in the transient regime and oscillations at later times. The ground truth is displayed in blue.

dynamics to learn from such a system are the population dynamics $\rho_{t,\mu}$ rather than the sample dynamics; see Appendix B.2. We observe the particles computed with a particle-in-cell method and learn the gradient field $\nabla s_{t,\mu}$ with the proposed HOAM approach. For a test physics parameter $\mu$ that controls the wave number, we then generate samples with $\nabla s_{t,\mu}$ and plot a histogram in Figure 3 for the bump-on-tail (top) and two-stream (middle top) instability. Our approach approximates well the histogram obtained with the classical particle-in-cell method. Figure 5 (right) shows that HOAM is the only method which provides speedup over the classical particle-in-cell (full) model, as NCSM and CFM lead to 1–2 orders of magnitude longer inference times than HOAM and the full models.

For the strong Landau damping problem in six dimensions (three spatial and three velocity), our HOAM approach achieves about 2 orders of magnitude speedup. This is because the runtime of the full model based on the traditional particle-in-cell method to compute the mean-field dynamics scales poorly with the dimension. In this example, the runtime of the full model increases by almost two orders of magnitude. In contrast, the runtime of our HOAM reduced model increases only from 6 to 8 seconds. This importantly shows that the computational costs of the inference step of reduced models built with HOAM avoid exponential scaling with the dimension in this example.

We now compute the electric energy as a quantity of interest from the generated samples over time $t$ for the test physics parameters, which we plot in Figure 4 and its relative error averaged over time (e.e.) in Table 1 (see (25)). Our HOAM approximates the electric energy well at later times, whereas NCSM and CFM lead to poorer approximations at later times $t$. This is relevant because this non-linear regime is where numerical solvers become important; the initial (linear) growth regime can be approximated well by analytical perturbation theory. Also for the six-dimensional strong Landau damping problem, our HOAM approach provides accurate predictions of the electric energy with orders of magnitude speedups; see Table 1 and Figure 3 as well as Figure 7 in the appendix.

**Speedups in inference step (predictions)** Recall two limitations of introducing a time and physics parameter dependence in NCSM/CFM via conditioning (see page 3 and Section 2.3): (i) For each $t$ and $\mu$, a separate sampling path has to be computed, which leads to orders of magnitude higher inference runtimes than in HOAM; see Table 1, Section 2.3. (ii) For each $t$ and $\mu$ pair, the target distribution $\rho_{t,\mu}$ is different, which can require $t$- and $\mu$-specific tuning of hyper-parameters of the inference step, which is impractical and thus can lead to a deterioration of accuracy compared to our HOAM approach; see Figure 3–4.

**Predicting statistics of chaotic and particle dynamics in high dimensions** We now consider the nine-dimensional dynamical system introduced in [69], which leads to chaotic behavior. We show in Figure 3 (bottom) the sample histogram corresponding to a test physics parameter that represents the Rayleigh number. At time $t = 3.7$ and projecting onto

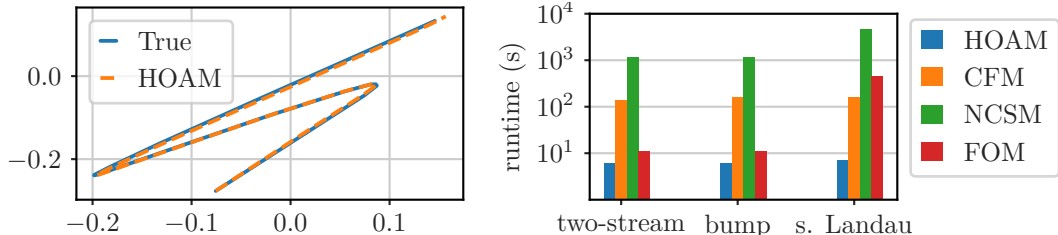

Figure 5: **Left**: HOAM accurately predicts the time evolution of the mean position of a 100-dimensional particle system in an aharmonic moving trap (dim 1 vs dim 100). **Right**: HOAM reduced models provide about 2 orders of magnitude speedup over traditional numerical (full) models for the 6 dimensional strong Landau problem. HOAM is also 1–2 orders of magnitude faster than CFM and NCSM, which provide no speedup over the full models in our problems.

| example: | two-stream | | bump-on-tail | | strong Landau | | 9D chaos | |
|---|---|---|---|---|---|---|---|---|
| metric: | e.e. | r.t. [s] | e.e. | r.t. [s] | e.e. | r.t. [s] | sinkhorn | r.t. [s] |
| CFM [2, 54] | 1.44 | 139 | 5.52 | 141 | 0.629 | 161 | 0.259 | 36 |
| NCSM [80] | 0.245 | 1142 | 0.626 | 1133 | 4.06 | 4531 | 0.869 | 1109 |
| AM [61] | 0.275 | 6 | 0.892 | 6 | NaN | - | 80.1 | 7 |
| HOAM-S (ours) | **0.078** | 6 | **0.427** | 6 | 0.641 | 7 | **0.214** | 7 |
| HOAM-G (ours) | 0.208 | 6 | 0.429 | 6 | **0.447** | 7 | 0.217 | 7 |

Table 1: HOAM with Simpson's and Gauss quadrature outperforms state-of-the-art methods w.r.t. inference runtime (r.t.) with comparable errors when applied to various physics problems for parametric model reduction. Metrics: e.e. is the relative error in electric energy, see (25); for the Sinkhorn divergence, see Appendix B.5.

dimension three and nine, the histograms show that the proposed HOAM accurately matches the low probability region that connects the two high probability regions, whereas AM fails to converge. Consider now the example of the particles in an aharmonic trap, which leads to 100-dimensional samples $X^i_{t,\mu}$. For a test physics parameter, Figure 5 shows that HOAM accurately predicts the mean particle positions even for this high dimensional system.

## 4 Conclusions, limitations, and future work

For parametric model reduction, learning population dynamics via minimal-energy vector fields over time $t$ and physics parameter $\mu$ with our variational approach helps reduce inference runtime compared to standard diffusion- and flow-based modeling that condition on $t$ and $\mu$ and therefore have to solve a separate inference problem for each time step and physics parameter at test time. Because we learn the dynamics over time $t$, it is critical to accurately capture the coupling over the time steps, for which we propose to use higher-order quadrature schemes when estimating time integrals in the training loss. The higher-order quadrature of the time integrals considerably improves training stability. Our approach achieves comparable errors as state-of-the-art methods while at the same time reducing inference runtime by 1–2 orders of magnitude. Additionally, HOAM provides speedups of up to 2 orders of magnitude to classical numerical full models.

*Limitations*: First, if there are only very few samples in time, even numerical quadrature cannot provide an accurate enough estimation of the loss, which could be a limitation in computational biology [23, 9]. Second, we currently seek a vector field that minimizes the kinetic energy or a variant thereof. Investigating other notions of energy that might lead to vector fields with other desired properties in certain problems remains a challenge.

We do not expect that this work has negative societal impacts.

## Acknowledgements

Berman and Peherstorfer were partially supported by the Air Force Office of Scientific Research under award FA9550-21-1-0222. Peherstorfer and Blickhan were partially supported by the Office of Naval Research, United States under award N00014-22-1-2728. We thank Stefan Possanner and Dominik Bell (Max Planck Institute for Plasma Physics) for their support with using the high-fidelity code used in the six-dimensional Vlasov experiments.

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

# A  Quadrature rules

## A.1  Monte Carlo estimation

Monte Carlo estimation approximates an integral by evaluating the integrand at randomly sampled nodes within an interval $[a, b]$,

$$\int_a^b f(t)\,dt \approx \frac{b-a}{N} \sum_{i=1}^{N} f(t_i). \tag{9}$$

We consider only the case where the nodes $t_i$ are uniformly distributed random variables in $[a, b]$. The weights are:

$$w_i = \frac{b-a}{N}, \quad i = 1, 2, \dots, N. \tag{10}$$

The root mean-squared integration error of Monte Carlo estimation decays as $\mathcal{O}(N^{-1/2})$, for integrands with bounded variance.

## A.2  Trapezoidal Rule

The trapezoidal rule approximates the integral of a function $f$ over an interval $[a, b]$ by dividing it into $N$ subintervals of equal width $h = \dfrac{b-a}{N}$. The approximation is given by:

$$\int_a^b f(t)\,dt \approx h \left[ \frac{1}{2}f(t_0) + \sum_{i=1}^{N-1} f(t_i) + \frac{1}{2}f(t_N) \right], \tag{11}$$

where the nodes $t_i$ are:

$$t_i = a + ih, \quad i = 0, 1, \dots, N. \tag{12}$$

The trapezoidal rule is a second-order rule, which means that the integration error decays as $\mathcal{O}(h^2)$ for sufficiently smooth functions.

## A.3  Composite Simpson's Rule

Composite Simpson's rule approximates the integral by fitting parabolas through intervals. It divides $[a, b]$ into an even number $N$ of subintervals of width $h = \dfrac{b-a}{N}$. The approximation is:

$$\int_a^b f(t)\,dt \approx \frac{h}{3} \left[ f(t_0) + 2 \sum_{i=1}^{N/2-1} f(t_{2i}) + 4 \sum_{i=1}^{N/2} f(t_{2i-1}) + f(t_N) \right], \tag{13}$$

with nodes:

$$t_i = a + ih, \quad i = 0, 1, \dots, N. \tag{14}$$

Composite Simpson's rule is a fourth-order rule, which means that the integration error decays as $\mathcal{O}(h^4)$ for sufficiently smooth functions.

## A.4  Gauss-Legendre Quadrature

Gauss-Legendre quadrature approximates the integral over $[-1, 1]$ by choosing nodes $t_i$ and weights $w_i$ so that polynomials of the highest possible degree are integrated exactly. A Gauss-Legendre quadrature has the form

$$\int_{-1}^{1} f(t)\,dt \approx \sum_{i=1}^{n} w_i f(t_i), \tag{15}$$

where $t_i$ are the roots of the Legendre polynomial $P_n(t)$, and the weights are:

$$w_i = \frac{2}{(1-t_i^2)[P_n'(t_i)]^2}. \tag{16}$$

For integration over $[a, b]$, a linear transformation maps $[-1, 1]$ to $[a, b]$:

$$\tilde{t}_i = \frac{b-a}{2}t_i + \frac{a+b}{2}, \quad \tilde{w}_i = \frac{b-a}{2}w_i. \tag{17}$$

The approximation becomes:

$$\int_a^b f(t)\,dt \approx \sum_{i=1}^n \tilde{w}_i f(\tilde{t}_i). \tag{18}$$

Gauss-Legendre quadrature exactly integrates polynomials of degree $2n - 1$, where $n$ is the number of nodes.

# B   Details about numerical examples

## B.1   Harmonic oscillator with background collisions

The equation of motion in four-dimensional phase-space for $X = [X_1, X_2, V_1, V_2]$ is given by:

$$\frac{\mathrm{d}}{\mathrm{d}t}\begin{bmatrix} X_1 \\ X_2 \\ V_1 \\ V_2 \end{bmatrix}(t) = \begin{bmatrix} V_1 \\ V_2 \\ -\omega^2 X_1 \\ -\omega^2 X_2 \end{bmatrix}(t) + \begin{bmatrix} 0 \\ 0 \\ \eta \\ \eta \end{bmatrix}\xi(t). \tag{19}$$

Here, $\eta = 5 \times 10^{-2}$ and $\xi$ denotes white noise. The initial configuration is a Gaussian centered at $m_0 = [1, 1]$ with covariance equal to $\Sigma_0 = 10^{-2} \times \mathrm{Id}$ in the spatial coordinates $X_1, X_2$ and Gaussian in the velocity coordinates $V_1, V_2$ centered at zero and with covariance $\Sigma_0 = 10^{-2} \times \mathrm{Id}$.

## B.2   Vlasov-Poisson problems

**Mean field approximations**  The Vlasov-Poisson system describes the interaction of charged particles. Due to the presence of the Coulomb force, the dynamics of a single particle depend on the position of all other particles. Assuming $N$ particles in the system, this means $\frac{\mathrm{d}}{\mathrm{d}t}X_{t,\mu}^i = v(t, X_{t,\mu}^i; \mu, X_{t,\mu}^1, \ldots, X_{t,\mu}^N)$. Given the fact that $N$ is in practice extremely large, it is natural to pass to the *mean-field limit*. Assuming the particles are indistinguishable, the result is a PDE of the form $\partial_t \rho_{t,\mu} + \nabla \cdot (\rho_{t,\mu} v_{\mathrm{mf}}(t, \cdot; \mu, \rho_{t,\mu})) = 0$ that describes the evolution of the collection (or population, ensemble) of particles denoted by $\rho_{t,\mu}$. In the specific case of the Vlasov-Poisson problem, Coulomb interactions in the mean-field limit give rise to a Poisson equation determining an electric field that is generated by the collection of particles and influences its dynamics. For completeness sake, we mention that the singularity of the Coulomb interaction poses a considerable technical challenge when passing to this limit. We refer to [53, 59] for the derivation of the Vlasov-Poisson equation and [60] for more examples of mean-field systems. The theory behind the test-cases we run in this work can be found in [57], Chapter 3.

**Governing equation**  We slightly change the notation here to be consistent with the references. $f : \mathcal{X}_x \times \mathbb{R}^d \times \mathbb{R} \times \mathcal{D} \to \mathbb{R}$, $d \in \{1, 2, 3\}$, denotes the distribution function governed by the Vlasov-Poisson system

$$\partial_t f(x, v, t; \mu) = -v \cdot \nabla_x f(x, v, t; \mu) - \nabla\phi(x, t) \cdot \nabla_v f(x, v, t; \mu) = 0, \tag{20}$$

$$-\mu^2 \Delta\phi(x, t; \mu) = 1 - \int_{\mathbb{R}^d} f(x, v, t; \mu)\mathrm{d}v. \tag{21}$$

In the notation of the rest of this work, $f(\cdot, \cdot, t; \mu) = \rho_{t,\mu}$, $\mathcal{X}_x \times \mathbb{R}^d = \mathcal{X}$. The spatial domain $\mathcal{X}_x$ is a subset of $\mathbb{R}^d$, in all our examples it is of the form $[0, l_1] \times [0, l_2] \times [0, l_3]$ with periodic boundary conditions.

**Two-stream instability** In this case, $d = 2$, so the particle positions vary in $\mathcal{X}_x = [0, l_1]$ with periodic boundary conditions and their velocity evolves in $\mathbb{R}$. For the two-stream instability, we set the initial distribution to

$$f_0(x,v) := \frac{1}{2\sqrt{2\pi}} \left( 1 + \alpha \cos \left( 2\pi \frac{x}{l_1} \right) \right) \left( \exp \left( -\frac{(v - v_0)^2}{2} \right) + \exp \left( -\frac{(v + v_0)^2}{2} \right) \right), \quad (22)$$

with $\alpha = 0.05, l_1 = 50, v_0 = 3$. The parameter $\mu$ varies as $\mu_{\text{train}} \in \{1.2, 1.3, \ldots, 1.9\}$ and $\mu_{\text{test}} \in \{1.25, 1.85\}$. We use a particle-in-cell method for generating the data based on the repository `https://github.com/pmocz/pic-python`. The number of marker particles is $N = 25000$ and for the sake of computing the electric field, a uniform grid of $N/8$ cells is used. Integration in time is done via a Störmer-Verlet splitting over $t \in [0, 40]$ with time-step size $10^{-2}$.

**Bump-on tail** We consider the initial distribution

$$f_0(x,v) = \frac{1}{\sqrt{2\pi}} \left( 1 + \alpha \cos \left( 2\pi \frac{x}{l_1} \right) \right) \left( \frac{\delta}{\sigma_1} \exp \left( -\frac{v^2}{2\sigma_1^2} \right) + \frac{1-\delta}{\sigma_2} \exp \left( -\frac{(v - v_b)^2}{2\sigma_2^2} \right) \right),$$
$$(23)$$

with $\alpha = 0.05, l_1 = 50, v_b = 4, \delta = 9/10, \sigma_1 = 1, \sigma_2 = 1/\sqrt{2}$. The parameter $\mu$ varies as $\mu_{\text{train}} \in \{1.3, 1.4, \ldots, 2.0\}$ and $\mu_{\text{test}} \in \{1.35, 1.95\}$. The other parameters are the same as in the two-stream case.

**Strong Landau damping** In this case, $d = 6$ and

$$f_0(x,v) = \frac{1}{\sqrt{2\pi}^3} \left( 1 + \alpha \cos \left( 2\pi \frac{x_1}{l_1} \right) \right) \exp \left( -\frac{|v|^2}{2} \right), \quad (24)$$

with $l_1 = 4\pi$ and $l_2 = l_3 = 1$. The data is generated using the Struphy package [67], the exact specifications of the simulation are available at `https://gitlab.mpcdf.mpg.de/struphy` as an example problem. The physics parameter we vary is the mass of the charged particles, which has the effect of changing the strength of the inertial term accelerating the particles relative to the advection term $v \cdot \nabla_x f$. This implies $\mu \in \{0.5, 0.6, \ldots, 1.5\}$, where $\mu = 1.0$ corresponds to the default settings. This $\mu = 1.0$ is also the test parameter and is excluded from the training set. The timing for the full order method has been obtained on a computing cluster with AMD EPYC Genoa 9554 CPUs using 8 MPI processes, which is a default option of the used code. For a single MPI process, it extends to 27 minutes while for 16, it can be reduced to 4 minutes.

The high-fidelity data we generate is using a control variate approach in order to reduce numerical noise introduced by the finite number of marker particles. Since we require the particles to be identical for our method, we assume they are all weighted equally when re-constructing the electric potential. This biases our reconstructed potential in comparison to the physical one, but we observe in practice that this is only by a multiplicative constant. We save $10^5$ marker particles from the high-order simulations and use $N = 25000$ of them as input data for our method. We integrate in time over $t \in [0, 8.75]$

## B.3 High-dimensional chaos

We consider the dynamical system introduced in [69]. We generate samples by initializing a 9 dimensional Gaussian centered at the origin with width equal to $2 \times 10^{-2}$. We then integrate these samples forward as an SDE whose drift is given by the 9-dimensional system of ODEs described in [69] and whose diffusion term is given as diagonal noise equal to $5 \times 10^{-2}$. We integrate 25000 particles of the system up to $T = 20$ using the Euler-Maruyama scheme with time step size equal to $10^{-2}$. The parameter $\mu$ varies as $\mu_{\text{train}} \in \{13.5, 13.6, \ldots, 14.2\}$ and $\mu_{\text{test}} \in \{13.65, 14.05\}$.

## B.4 Particles in aharmonic trap

We consider the evolution of interacting particles in an aharmonic trap [16]. The two-dimensional particle positions $Z_1(t, \mu), \ldots, Z_M(t, \mu)$ are governed by an SDE

$$\mathrm{d}Z_i = g(t, Z_i)\mathrm{d}t + \sum_{j=1}^{M} K(Z_i, Z_j)\mathrm{d}t + \sqrt{2\gamma}\mathrm{d}W_i, \qquad i = 1, \ldots, M,$$

where $\gamma > 0$ is the diffusion coefficient and $W_i$ are independent Wiener processes. The function $g(t, Z) = (a(t) - Z)^3$ describes a time-dependent one-body force, where $a(t) = 5/4(\sin(\pi t) + 3/2)) + \mu \cos(2\pi t)$ is the position of the trap. The function $K(Z, Z') = \frac{\alpha}{M}(Z' - Z)$ describes a pairwise interaction term. We set $\alpha = -1/4$ and $\gamma = 10^{-2}$. The parameter $\mu$ is in the range $\mathcal{D} = [0.3, 0.9]$ and modifies the position of the trap. A sample $X_{t,\mu}^i$ corresponds to a vector $[Z_1(t, \mu), \ldots, Z_M(t, \mu)]^T$ of dimension 100, because we have $M = 50$ particles and each position $Z_j(t, \mu)$ as two dimensions. We generate samples via Monte Carlo by using the Euler-Maruyama scheme. The time step size is $\delta t = 10^{-3}$ and we integrate up to final time 2.

## B.5 Other details about numerical experiments

In terms of network architecture, we follow [13] closely because we use their network architecture. We use MLPs to parameterize both the main network and the hyper-network with swish activation functions. The main network is depth 7 and width 64 linear layers while the hyper-network is depth 3 with width 15 linear layers. The rank of the CoLoRA modulations is set to 3. Identical CoLoRA architectures are used for all HOAM experiments as well as the comparisons with AM, NCSM, and CFM. The only difference is the size of the output layer for NCSM and CFM whose outputs must be the same dimensionality as their inputs.

For all experiments we use an Adam optimizer at a $2 \times 10^{-3}$ learning rate with a cosine learning rate scheduler. For all experiments unless otherwise noted, we take a batch size of 256 particles over 256 time points. We optimize for $50,000$ Adam iterations for Vlasov examples and for $25,000$ Adam iterations for all other examples.

The results were computed on NVIDIA Quadro RTX 8000 GPUs. All code was implemented in Python using the JAX library with JIT complication where possible.

Hyper-parameter $\epsilon$ in the loss (7) searched over $\{0, 1, 2, 5, 7\} \times 10^{-2}$ for both HOAM and AM.

The relative error in the electric energy is computed as

$$\frac{1}{T} \sum_{t=1}^{T} \frac{|e_{\mathrm{true}}(t) - e_{\mathrm{predict}}(t)|}{|e_{\mathrm{true}}(t)|}, \tag{25}$$

where $e_{\mathrm{true}}(t)$ is the electric energy predicted by the high-fidelity numerical simulations at time $t$ and $e_{\mathrm{predict}}(t)$ is the electric energy computed from samples of either HOAM (ours), AM, NCSM, or CFM. The relative error in the mean is

$$\frac{1}{T} \sum_{t=1}^{T} \frac{|\mathbb{E}[\rho_{\mathrm{true}}(t)] - \mathbb{E}[\rho_{\mathrm{predict}}(t)]|}{|\mathbb{E}[\rho_{\mathrm{true}}(t)]|}, \tag{26}$$

where the expected values are estimated via Monte Carlo from the generated samples.

The Sinkhorn distance is computed with `https://ott-jax.readthedocs.io/en/latest/` with threshold $10^{-3}$; see also [25].

## B.6 Additional numerical results

In Figure 6 we show the various projections at time $t = 3.7$ of the sample distribution corresponding to the nine-dimensional chaotic system [69].

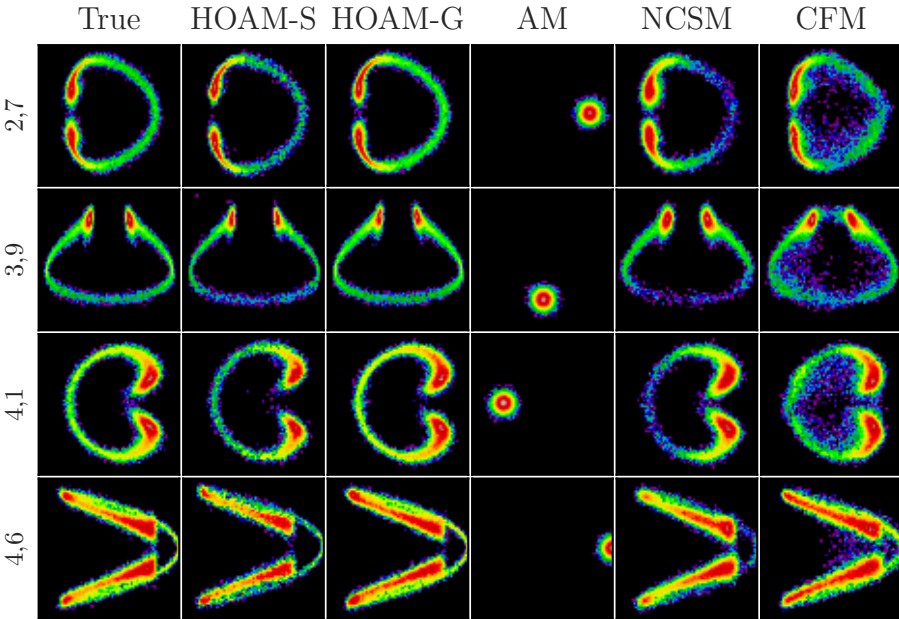

Figure 6: Shows the projections of other dimensions of the nine-dimensional chaotic system [69]; see also Figure 3.

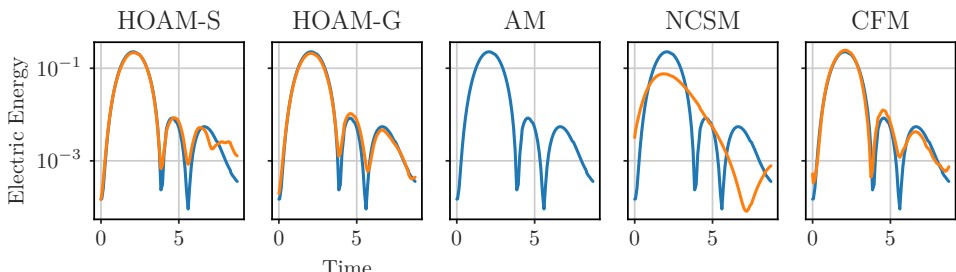

Figure 7: Electric energy and solution field at time $t = 4$ for the 6 dimensional strong Landau example.

In Figure 7 we show the particle histograms and the electric energy curves for the six-dimensional Vlasov-Poisson problem corresponding to strong Landau damping.

In Figure 8, for the linear oscillator example, we compare CoLoRA to two other modulation schemes: FiLM [66] and MLP. For the MLP the inputs $x, t, \mu$ are concatenated together and input directly to the model. There is no hyper-network or modulation scheme. For FiLM, we closely follow the original paper. The main network takes $x$ as input and the hyper-network $t, \mu$ as input. The hyper-network and main network have the same parameter counts as in the CoLoRA experiments. The output of the hyper-network then directly modulates the activation of each layer of the main network as detailed in the original FiLM paper [66]. Figure 8 shows that parameterizing the vector field $s_{t,\mu}$ with CoLoRA layers achieves the lowest mean Wasserstein distance, which motivates the use of the CoLoRA modulation scheme [13].

## C  Calculations regarding the entropic loss

In the following, assume that $\rho \in \mathcal{P}(\mathcal{X})$ is a smooth density bounded away from zero. We begin by showing some calculation rules of the operator $-\Delta_\rho : s \mapsto -\nabla \cdot (\rho \nabla s)$ with

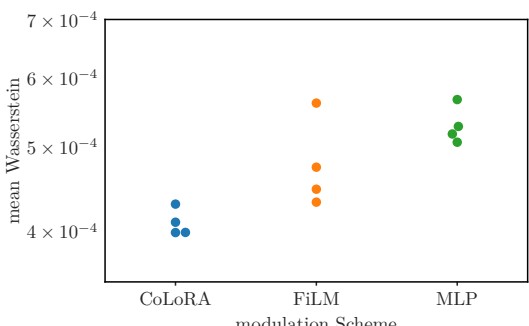

Figure 8: Comparison of CoLoRA modulation scheme [13] versus FiLM [66] and MLP. CoLoRA layers achieve the lowest mean Wasserstein distance compared to FiLM and MLP. In particular, CoLoRA avoids outliers with larger errors.

homgeneous Neumann boundary conditions. In its weak form, it reads

$$-\int_{\mathcal{X}} f \Delta_\rho s \mathrm{d}x = \int_{\mathcal{X}} \nabla f \cdot \nabla s \, \rho \mathrm{d}x \quad \forall f \in \mathcal{C}^\infty(\mathcal{X}). \tag{27}$$

With the choice $f = \log \rho$, we find the useful identity $\Delta_\rho \log \rho = \Delta \rho$. Next, recall the objective $E^\epsilon$ from (5):

$$E^\varepsilon(s) = \frac{1}{2} \int_{\mathcal{X}} \left| \nabla \left( s - \frac{\varepsilon^2}{2} \log \rho \right) \right|^2 \rho \mathrm{d}x - \int_{\mathcal{X}} \partial_t \rho s \mathrm{d}x.$$

Now denote by $\delta s$ an arbitrary element of $\mathcal{C}^\infty(\mathcal{X})$. Then, if $s^\epsilon$ is a minimizer of the (strictly convex) objective, it holds that

$$0 \overset{!}{=} \frac{\mathrm{d}}{\mathrm{d}\tau} E^\varepsilon(s^\epsilon + \tau \delta s)\Big|_{\tau=0} = -\int_{\mathcal{X}} \delta s \Delta_\rho \left( s^\epsilon - \frac{\varepsilon^2}{2} \log \rho \right) \mathrm{d}x - \int_{\mathcal{X}} \partial_t \rho \delta s \mathrm{d}x \quad \forall \delta s. \tag{28}$$

Hence,

$$0 = \Delta_\rho \left( s^\epsilon - \frac{\varepsilon^2}{2} \log \rho \right) + \partial_t \rho = \nabla \cdot (\rho \nabla s^\epsilon) - \frac{\varepsilon^2}{2} \Delta \rho + \partial_t \rho. \tag{29}$$

Furthermore, note that (5) is identical to

$$E_{t,\mu}^\epsilon(s) = \int_{\mathcal{X}} \left( \frac{1}{2} |\nabla s|^2 + \frac{\epsilon^2}{2} \Delta s \right) \rho_{t,\mu} \mathrm{d}x - \int_{\mathcal{X}} \partial_t \rho_{t,\mu} s \mathrm{d}x + \frac{\epsilon^2}{8} \int_{\mathcal{X}} |\nabla \log \rho_{t,\mu}|^2 \rho_{t,\mu} \mathrm{d}x \tag{30}$$

after integration by parts. The last term is the Fisher information of the data at $t, \mu$ and plays no role in the optimization.

## D  Motivating the partial integration in time in the loss

Note that the problems from Equation (3) corresponding to different values of $t$ are coupled through the term $\partial_t \rho_{t,\mu}$. This is most apparent when one discretizes the equation in time. Denote by $\{t_i\}_{i=0}^{n_t}$ a strictly increasing sequence with $t_0 = 0, t_{n_t} = 1$, and $t_{i+1} - t_i = \delta t_i$. Then, for fixed but arbitrary $\mu$, we obtain $n_t$ coupled problems of the form

$$\min_{s_{t_i} \in H^1(\rho_{t_i,\mu}, \mathcal{X})} \frac{1}{2} \int_{\mathcal{X}} |\nabla s_{t_i,\mu}|^2 \rho_{t_i,\mu} \mathrm{d}x - \frac{1}{\delta t} \int_{\mathcal{X}} (\rho_{t_{i+1},\mu} - \rho_{t_i,\mu}) s_{t_i,\mu} \mathrm{d}x \quad \forall i, \mu. \tag{31}$$

Adding these problems and shifting the indices, one can eliminate $\rho_{t_{i+1},\mu}$, explicitly coupling $s_{t_i,\mu}$ and $s_{t_{i+1},\mu}$. The continuous equivalent of this of course is an integration over $t$, followed by an integration by parts.

# E   Geometric picture of the optimization problem

We omit the dependence on the parameter $\mu$ here for the sake of simpler notation and write $d\rho$ for $\rho\,dx$ for brevity. Note that the following considerations are purely formal. They are meant to illustrate a geometric picture of the optimization problems we consider. We claim no originality of these ideas; the exposition is based on Chapter 7 of [89] as well as [24, 55].

**Otto calculus**   Based on the identification of the tangent space of $P(\mathcal{X})$ with the space of gradients (more rigorously, at point $\rho_t \in P(\mathcal{X})$, the closure of $\{\nabla f : f \in \mathcal{C}^\infty(\mathcal{X})\}$ in $L^2(\mathcal{X}, \rho_t, \mathbb{R}^d)$, see Definition 8.4.1 in [5]), one can view $\mathcal{P}(\mathcal{X})$ formally as a Riemannian manifold:

**Definition 1** ([63]). *Let $\tau \mapsto \rho_\tau^1$ and $\tau \mapsto \rho_\tau^2$ be two curves valued in $\mathcal{P}(\mathcal{X})$ for $\tau \in (t-\epsilon, t+\epsilon)$ such that $\rho_\tau^1\big|_{\tau=t} = \rho_\tau^2\big|_{\tau=t} = \rho_t$. The optimal transport metric on $T\mathcal{P}(\mathcal{X})$ at $\rho_t \in P(\mathcal{X})$ is given by*

$$g(\rho_t)(\partial_\tau \rho_\tau^1\big|_{\tau=t}, \partial_\tau \rho_\tau^2\big|_{\tau=t}) = \int_{\mathcal{X}} (\nabla s_t^1 \cdot \nabla s_t^2) d\rho_t :$$
$$\partial_\tau \rho_\tau^1 + \nabla \cdot (\rho_t \nabla s_t^1) = 0, \partial_\tau \rho_\tau^2 + \nabla \cdot (\rho_t \nabla s_t^2) = 0. \quad (32)$$

This formalism is commonly named after the author of [63] and is closely linked to Arnold's considerations on geometric hydrodynamics [6][2] As both the identification of $s_t$ from $\partial_t \rho_t$ and the metric depend on $\rho_t$, the geometry defined on $\mathcal{P}(\mathcal{X})$ in this way is non-trivial.

**Action of a curve**   The optimization Equation (3) has an appealing physical interpretation: The vector field we define as tangent to the curve is, among all compatible ones, the one with the smallest integrated kinetic energy. In analogy with the physical literature, we call $\frac{1}{2}\int_0^1 \int_{\mathcal{X}} |\nabla s_t|^2 d\rho_t$ the *action* of the curve $t \mapsto \rho_t$ with tangent velocity $\nabla s_t$. We want to stress that while this procedure is reminiscent of physical action principles, in the latter a solution corresponds to a *stationary point* given boundary conditions at the beginning and end of the curve. The problem we consider in Equation (6) is more narrow and concerned with finding $\nabla s_t$ that matches a *given* curve $t \mapsto \rho_t$. Determining curves of minimal action in $\mathcal{P}(\mathcal{X})$, leads to the Benamou-Brenier formula ([4], Proposition 2.30):

$$\frac{1}{2}W_2^2(\rho_0, \rho_1) = \inf_{\rho,s} \left( \frac{1}{2} \int_0^1 \int_{\mathcal{X}} |\nabla s_t|^2 d\rho_t\, dt : \partial_t \rho_t + \nabla \cdot (\rho_t \nabla s_t) = 0, \rho_{t=0} = \rho_0, \rho_{t=1} = \rho_1 \right),$$
$$(33)$$

with $W_2$ the Wasserstein (or Kantorochiv-Rubinstein) distance.

**Lagrangian functions**   The selection criterion based on kinetic energy alone is not without alternatives. In [24], the relation $\partial_t \rho = -\Delta_\rho s$ is interpreted as a form of Legendre transform, hence $s$ plays the role of a momentum and $L(\rho_t, \partial_t \rho_t, t) = \int_{\mathcal{X}} |\nabla \Delta_\rho^\dagger \partial_t \rho|^2 d\rho$ that of a Lagrangian. Here, we introduced the notation $\Delta_\rho^\dagger$ to denote the pseudo inverse operator. Note that, formally, it is sensible to consider $\partial_t \rho$ as an element of the tangent space of $\mathcal{P}(\mathcal{X})$. After all, $\rho + \tau \partial_t \rho \in \mathcal{P}(\mathcal{X})$ for $\rho$ strictly positive and $\tau$ small enough. In this picture, $s$ is an element of the cotangent space. The introduction of [63] addresses the two concepts and how they relate.

Any function $L : (\rho, \partial_t \rho, t) \mapsto L(\rho, \partial_t \rho, t)$, strictly convex and superlinear in its second argument, can be chosen to define the minimization objective.[3]  Details can be found in Chapter 7 of [89], which also features a comprehensive discussion of the history and applications of this problem. In recent years, this formulation has been applied for modeling purposes, e.g. in [44]. To give an example, the choice $L(\rho, \partial_t \rho, t) = \frac{1}{2} \int_{\mathcal{X}} |\nabla \Delta_\rho^\dagger \partial_t \rho|^2 d\rho - \int_{\mathcal{X}} V d\rho$ for a potential $V : \mathcal{X} \to \mathbb{R}$ can be used to model obstacles in the path of the samples.

---

[2]The derivation of fluid dynamics from variational principles is, of course, much older and goes back as far as Langrange's Mécanique analytique published in 1789.

[3]The variables $\rho$ and $\partial_t$ here denote any probability density and a scalar field on $\mathcal{X}$.

There exist a number of partial differential equations whose solutions $\rho_t$ can be described as curves of stationary action with respect to such Lagrangians, described in [3, 24], as well as [89], Chapter 23, and [90], Chapter 8.

**Schrödinger Bridge**   The objective defined in Equation (5) corresponds to the choice

$$L^\epsilon(\rho, \partial_t \rho, t) := \frac{1}{2} \int_{\mathcal{X}} \left| \nabla \left( -\Delta_\rho^\dagger \partial_t \rho + \frac{\epsilon^2}{2} \log \rho \right) \right|^2 \mathrm{d}\rho. \tag{34}$$

The associated momentum $s^\epsilon$ therefore satisfies $s^\epsilon = \frac{\delta L^\epsilon}{\delta(\partial_t \rho)}$, hence $-\Delta_\rho s^\epsilon + \frac{\epsilon^2}{2} \Delta \rho = \partial_t \rho$, a Fokker-Planck equation. Furthermore, the action of the curve $t \mapsto \rho_t$ is given by

$$\int_0^1 L^\epsilon(\rho_t, \partial_t \rho, t) \mathrm{d}t = \int_0^1 \left( \frac{1}{2} \int_{\mathcal{X}} |\nabla \Delta_\rho^\dagger \partial_t \rho|^2 \mathrm{d}\rho + \frac{\epsilon^4}{8} \int_{\mathcal{X}} |\nabla \log \rho_t|^2 \, d\rho_t \right) dt$$
$$+ \frac{\epsilon^2}{2} \left( \int_{\mathcal{X}} \log \rho_t \mathrm{d}\rho_t \Big|_{t=1} - \int_{\mathcal{X}} \log \rho_t \mathrm{d}\rho_t \Big|_{t=0} \right). \tag{35}$$

This expression is known as the dual formulation of the Kantorovich-Schrödinger problem ([31], Theorem 36, except for the fact that the $\epsilon$ therein corresponds to $\epsilon^2/2$ here). While the classical optimal transport problem is concerned with the path connecting $\rho_0$ and $\rho_1$ minimizing the time integral of the kinetic energy (which coincides with the transport cost), the Schrödinger-Bridge problem is concerned with finding the most likely configuration at intermediate times, subject to the information that the configuration is given at times 0 and 1 and assuming that the particles $X_t$ undergo Brownian motion with diffusivity $\varepsilon^2/2$. Unless $\rho_1$ is the result of a convolution of $\rho_0$ with a Gaussian kernel of width $\varepsilon$, the evolution of the system towards $\rho_1$ is a rare event and the most likely solution is to be understood conditional on the observation of this event.

Rigorous results can be found in Section 5 of [31]. Another derivation of the loss function from Equation (5), starting from the static formulation and linking to the dynamical picture presented here, can also be found in [47], Theorem 2.1. In their notation, $\Psi = -s$.

