# OpenReview forum: "Parametric model reduction of mean-field and stochastic systems via higher-order action matching"
_NeurIPS.cc/2024/Conference — NeurIPS 2024 poster_

### Official Review · Reviewer_rAtu · 2024-07-06

**Soundness:** 3
**Presentation:** 2
**Contribution:** 3
**Rating:** 5
**Confidence:** 4

**Summary:**

The authors present a framework for learning a time-dependent gradient vector field (parametric model) that describes how the particle evolves in the state space. The idea is based on the action-matching framework (https://arxiv.org/abs/2210.06662), which learns a time-dependent gradient field that interpolates between marginals at different times in a simulation-free manner. The authors propose using high-order quadrature rules for evaluating nested integrals in the objective. They demonstrate the forecasting ability of their framework on diverse examples in high dimensions and show that it outperforms other diffusion-based and flow-based models in inference runtime and accuracy

**Strengths:**

Developing efficient and high-fidelity surrogate models is of great importance to the computational science and engineering community. The authors take a step forward in this direction by learning the effective population dynamics of the underlying physical process. The paper is well-written, and the related works are adequately referenced.

**Weaknesses:**

The objective used by the authors is based on the action-matching paper, now adapted to include the dependence on the parameters (\mu). From a methodological standpoint, the authors' only important contribution is using a higher-order quadrature scheme to discretize the nested integrals

**Questions:**

General comments:

i) Maybe, I am not understanding things here.. Can the authors elaborate a bit more on the O(K\tau) complexity? If for instance, I train a velocity field using continuous normalizing Flow (simulation-based) using empirical marginals at different times. I can then integrate the learned velocity in time to get samples, in the same way, the authors draw samples from the gradient field through Langevin dynamics. Where is the extra \tau in complexity coming from?

ii) Or, is this framework a simulation-free strategy to train a Continuous Normalizing Flow but adapted for multiple snapshots? This was not so clear in the paper.

iii) The authors learn a gradient field as a function of parameters and time. Can I authors comment a bit on the stability of the learned velocity during inference time?

Specific comments

i) In Figure 5, a color bar showing the scales would help with the interpretation.

ii) In Figure 4, a label to show what the two curves (blue and orange) are is needed. What is the ground truth and what is the prediction?

**Limitations:**

-

---

> ### Author Rebuttal · Authors · 2024-08-06
>
> We thank the reviewer for their thorough and insightful comments:
>
> "The objective used by the authors is based on the action-matching paper, now adapted to include the dependence on the parameters (\mu) [...] the authors' only important contribution is using a higher-order quadrature scheme to discretize the nested integrals."
>
> - We stress that the higher-order quadrature scheme is not just a nice add-on to improve the accuracy a bit but the **core component to make the approach work in practice at all**. In fact, we show that action matching is unstable even on simple problems (Figure 2, see HOAM vs rand). That AM is unstable on presumably simple problems is also documented in community implementations such as the TorchCFM library. Moreover we show that AM fails on systems with complex dynamics and high dimensions (Strong Landau and 9D chaos, see Figure 8 and 9 in appendix). By contrast HOAM succeeds, allowing us to derive predictive surrogate models with low inference costs of such challenging problems for the first time.
> - We further uploaded a PDF that shows more detailed results showing that HOAM is essential for stable training as the method proposed in the action-matching framework fails on the problems we consider (see global response).
> - In terms of novelty of the objective function, variants of it have been widely used in other works far earlier than the action-matching paper (this is also stated in the AM paper). For example, there is [Reich 2011, page 240], with the earliest form appearing in [Otto & Villani 2000, Section 3, and Bennamou & Brenier 2000, eq. 35]. Thus, rather than inventing a new objective, we made the objective computationally tractable, which we consider an important contribution.
>
> \
> "Can the authors elaborate a bit more on the O(K\tau) complexity? [...] Where is the extra \tau in complexity coming from?"
> - The standard way of handling a time $t$ and parameter $\mu$ dependency with NCSM and CFM is conditioning (see literature review in paper). As we point out, this means that for each of the K time steps, a separate inference problem has to be solved, which is expensive: $\tau$ refers to the number of steps taken in solving the SDE/ODE in one inference step (for one out of the K time steps) in conditioned CFM/NCSM. Thus with conditioned CFM and NCSM a separate SDE must be solved at every $t$ and $\mu$ for which one wants samples.
> - By contrast, $\tau$ does not appear in the complexity for HOAM because $\nabla s$ evolves particles such that they match $\rho$ at each time. Thus physical time $t$ and the SDE time $\tau$ are aligned.
>
> \
> "[...] is this framework a simulation-free strategy to train a Continuous Normalizing Flow but adapted for multiple snapshots?"
> - As we discuss above, simple conditioning is not competitive for the surrogate modeling task that we consider. It is absolutely possible that other methods for training CNFs can be extended and modified to be more efficient for surrogate modeling, but instead of tweaking another method we opted to go with our approach as it is simulation-free and naturally couples the physical time t with the sampling time so that in one inference step a whole sample trajectory is obtained. This is what ultimately provides speedups, which are key for surrogate modeling.
>
> \
> "The authors learn a gradient field as a function of parameters and time. Can I authors comment a bit on the stability of the learned velocity during inference time?"
> - It is sufficient that the drift and diffusion are uniformly Lipschitz continuous such that the inference SDE (or ODE when $\epsilon = 0$) is well-posed. See, for example, [Ambrosio, Gigli & Savaré 2005, Lemma 8.1.4]. The diffusion is a constant in this work and because we parametrize with a CoLoRA neural network, the gradient $\nabla s$ is smooth in $t, x$, which is sufficient for the well-posedness. We will add comments about this in the paper if it gets accepted.
> - Regarding the analytical vector field we hope to converge to, its properties depend on the data. When $t \mapsto \rho_{t, \mu}$ describes a regular curve in Wasserstein space [Gigli 2012, Definition 2.7], then the inference ODE is well-posed [Gigli 2012, Theorem 2.6].
>
> \
> Specific comments about improving figures:
> - We will address these, if the paper gets accepted.
> - The ground truth in Figure 4 is blue.
>
> References:
> - [Ambrosio, Gigli & Savaré 2005] Luigi Ambrosio, Nicola Gigli, and Guiseppe Savaré. Gradient Flows. Lectures in Mathematics ETH Zürich. Birkhäuser-Verlag, Basel, 2005. doi:10.1007/978-3-7643-8722-8
> - [Bennamou & Brenier 2000] J.-D. Benamou and Y. Brenier. A computational fluid mechanics solution to the Monge-Kantorovich mass transfer problem. Numerische Mathematik, 84(3):375–393, Jan. 2000. doi:10.1007/s002110050002
> - [Gigli 2012] Nicola Gigli. Second Order Analysis on (P2(M),W2). Memoirs of the American Mathematical Society, Volume 216; 2012
> - [Otto & Villani 2000] F. Otto and C. Villani. Generalization of an Inequality by Talagrand and Links with the Logarithmic Sobolev Inequality. Journal of Functional Analysis, 173(2):361–400, June 2000. doi:10.1006/jfan.1999.3557
> - [Reich 2011] Reich, S. A dynamical systems framework for intermittent data assimilation. Bit Numer Math 51, 235–249 (2011). doi:10.1007/s10543-010-0302-4

---

> > ### Comment · Reviewer_rAtu · 2024-08-12
> >
> > I thank the authors for their detailed responses and clarification. I will retain my rating.

---

> > > ### Author Response · Authors · 2024-08-13
> > >
> > > Thank you for the comments. If there is any other information we can provide, please let us know.

---

### Official Review · Reviewer_gej1 · 2024-07-09

**Soundness:** 4
**Presentation:** 4
**Contribution:** 4
**Rating:** 9
**Confidence:** 4

**Summary:**

The authors focus on learning models for population dynamics of parameterized physical systems that exhibit stochastic and mean-field effects in time. To do so, the authors leverage the Benamou-Brenier formula to learn gradient fields that transport the probability density as time evolves, and that enable the generation of sample trajectories reflecting the dynamics of the population. Numerical experiments show compeling results and state-of-the-art performance in high-dimensional particle systems and in chaotic systems.

**Strengths:**

The contributions of the paper are novel, and the presentation is very clear. I commend the author's attempts at making the paper reasonably self-contained by adding context in Appendices C through E. Each of the theoretical developments are clearly motivated. Finding a vector field as proposed enables the interpolation of probability measures at different times. This notion has broad impacts beyond those stated in the paper. It is also interesting that the authors illustrate the importance of using a proper quadrature in time as opposed to random sampling. The experiments are also compelling, in particular the ability to resolve the low probability connection between two high probability regions as shown in Fig. 5.

**Weaknesses:**

To this reviewer the paper does not have any evident weakness. However, when introducing a high-order quadrature in time, which improves the performance, the authors do not elaborate on why one would, or would not, expect further improvements if similar quadrature rules were used in the spatial variable or in the physical parameters.

**Questions:**

I have the following minor questions:
- In (6) it seems that the integrand should be evaluated on $s_t$ instead of $s$ for consistency with (4).
- In (7) there seems to be a parenthesis missing that factors out the density $\rho_{t,\mu}$.

**Limitations:**

The authors adequately address the limitations of their work in Section 4. These are mostly technical. For instance, they require a high sampling rate in time, which is not always available in practice. This limits the applicability of the method, but does not detract from their main contribution.

---

> ### Author Rebuttal · Authors · 2024-08-06
>
> We thank the reviewer for the positive and thorough review. We will address all the points brought up.
>
> "In (6) it seems that the integrand should be evaluated on  instead of for consistency with (4)."
> - Thank you, yes, this should be $s_{t, \mu}$.
>
> "In (7) there seems to be a parenthesis missing that factors out the density rho"
> - Thank you, we will fix this in the final version, if the paper gets accepted.
>
> "the authors do not elaborate on why one would, or would not, expect further improvements if similar quadrature rules were used in the spatial variable or in the physical parameters."
> - Typically the dimension of $x$ is too high for quadrature rules to be effective, which is why we use Monte Carlo for it. We also specifically assume that our data is in the form of samples, so evaluating $\rho$ at quadrature points would require some form of density estimation, a non-trivial task in high dimensions.
> - For the parameter $\mu$, higher-order quadrature rules can be used but in surrogate modeling we typically have data points only at very few $\mu$ training parameters, which makes higher-order quadrature rules difficult to apply.

---

> > ### Comment · Reviewer_gej1 · 2024-08-13
> >
> > I thank the authors for their responses. I have the following additional question. In the pdf file you attached, Fig. 3 shows the relative error in mean and the caption states that "we see that HOAM remains relative stable, while AM increases in error." However, HOAM is the _blue_ line which not only is above the orange line (representing AM) but also seems to increase faster as $T$ increases. Is this a typo?

---

> > > ### Author Response · Authors · 2024-08-13
> > >
> > > Yes this is a typo. We thank the reviewer for catching this. We apologize for the confusion and will update the pdf accordingly.
> > >
> > > In all cases HOAM does outperform AM.

---

> > > > ### Comment · Reviewer_gej1 · 2024-08-13
> > > >
> > > > Thanks for the clarification.

---

### Official Review · Reviewer_zRLH · 2024-07-13

**Soundness:** 2
**Presentation:** 2
**Contribution:** 2
**Rating:** 4
**Confidence:** 3

**Summary:**

This paper develops models of population dynamics in physical systems that exhibit stochastic and mean-field effects, influenced by physics parameters. The goal is to create models that can efficiently predict system behavior as alternatives to classical numerical methods. By utilizing the Benamou-Brenier formula from optimal transport and action matching, the approach involves solving a variational problem to infer gradient fields that approximate population dynamics. These gradient fields enable the generation of sample trajectories that mimic physical system dynamics under various physics parameters. The study highlights the importance of combining Monte Carlo sampling with higher-order quadrature rules for accurate estimation and stable training. The models are demonstrated to perform well on Vlasov-Poisson instabilities and high-dimensional particle and chaotic systems, outperforming state-of-the-art diffusion-based and flow-based models that rely solely on time and physics parameters.

**Strengths:**

1)	Efficient parametric model reduction: The model is demonstrated to reduce inference runtime significantly compared to standard diffusion- and flow-based models by leveraging minimal-energy vector fields.
2)	Accurate and stable dynamics learning: It captures the coupling over time steps accurately, using higher-order quadrature schemes for estimating time integrals, which enhances training stability.
3)	High accuracy and reduced runtime: Achieves error rates comparable to state-of-the-art methods while reducing inference runtime by 1-2 orders of magnitude.

**Weaknesses:**

1) The model assumes access to a dense set of time points for the Gauss-Legendre quadrature, which may not be applicable when only a few time samples are available. This was already mentioned as a limitation of the current work. As this forms the main part of the approach, the practical benefit would be limited.
2) Regarding the vector field complexity, the model seeks a vector field that minimizes kinetic energy, but in some cases, this may be more complicated than other vector fields that produce the same population dynamics. Examples include situations where the minimal-energy field varies with time, making it challenging to determine the appropriate energies to use for different problems.

**Questions:**

1)	Authors aims to learn population dynamics $\rho_{t,\mu}$ instead of learning the dynamics of individual trajectories $t\to X_{t,\mu}^i$ for all $i$/. What are the assumptions on model so that it admits density? If not, would this work apply to the situation under weak convergence where no density is assumed. Moreover, it is not clear why the equations (5) to (7) should admit solutions in the strong forms.
2)	Parametrizing the $s_{t,\mu}$ with weight modulation as in CoLoRA [38] only applies to deterministic time-dependent dynamical systems. However, one goal of the work was stated as learning the population dynamics should allow for seamless treatment on deterministic and stochastic systems. How the latter can be handled with the corresponding parametrization is unclear. The forms of the layers assumes some low-rank structure, and only the weight modulations $\phi$ is assumed to depend on time and parameters.  As such, are stochastic systems omitted from the problem definition unlike the initial motivation?
3)	Solving eq (7) from data is challenging and prone to potential numerical issues. As a remedy, a combination of higher order numerical quadrature and MC sampling strategy is proposed. However, the details of this approach would be better to provide with a stability analysis on the training as mentioned in the text.

**Limitations:**

Yes, limitations of their work is adequately addressed. No potential negative societal impact of their work is identified.

---

> ### Author Rebuttal · Authors · 2024-08-06
>
> We thank the reviewer for the comments and respond to the following points:
>
> "The model assumes access to a dense set of time points for the Gauss-Legendre quadrature [...] the practical benefit would be limited"
> - Having data at a good amount of time points is a common situation in the setting of surrogate modeling and model reduction, which we consider here. The data are generated with a high-fidelity numerical model, which has to take small time steps for numerical stability reasons. With HOAM we show that not only we can leverage the dense sampling in time, but that it is absolutely critical to exploit it with appropriate numerical quadrature as existing methods fail due to inaccurate estimates of the time integral (see Figure 2 in paper as well as new results to provide further evidence in the uploaded PDF).
> - We stress that the problems that we examine here are real problems from the physics literature (see references) that are intractable with current surrogate modeling techniques (see references in paper). With HOAM we construct fast, predictive surrogate models for such problems for the first time. Additionally we show that other reasonable approaches like NCSM or CFM perform worse while incurring orders of magnitude higher inference costs. Thus, this shows that there is a large practical benefit to HOAM—creating fast surrogate models of stochastic and mean-field systems.
> - If the paper gets accepted, we will emphasize more clearly in the Introduction that in our problem setting it can be reasonably expected that data are available at many time points.
>
> \
> "[the learned vector fields] may be more complicated than other vector fields that produce the same population dynamics."
> - This is an interesting point, which we briefly discuss in the Conclusion section.
> - The main challenge is mathematically formalizing what “complicated” and “easy” means in the context of neural network approximations, which will also depend on the problem at hand.
> - We build on the kinetic energy because our results show it is a reasonable energy for many different problems that also links nicely with the theory developed in optimal transport theory (see our appendix). One desirable feature of the kinetic energy is that $\nabla s$ of the minimum kinetic energy vector field is identically zero when the population dynamics are stationary - this means that if the population dynamics are stationary, then sampling with the minimum kinetic energy field is trivial as the samples don’t move. This is not necessarily the case for other compatible vector fields.
> - If this paper gets accepted, we will expand the comments in the Conclusion section where we briefly discuss this point.
>
> \
> "Authors aim to learn population dynamics [...] What are the assumptions on the model so that it admits density? If not, would this work apply to the situation under weak convergence where no density is assumed. Moreover, it is not clear why the equations (5) to (7) should admit solutions in the strong forms."
> - Objective (7) (and (5), the special case where $\varepsilon = 0$) is expressed entirely in the form of expectations values w.r.t. $\rho$ - both are meaningful if no density is admitted and can be evaluated on empirical distributions (sums of dirac masses). We will add a comment about this if the paper gets accepted.
> - We use (5) and (7) to formally derive an optimality criterion for the learned gradient field $\nabla s^*(\theta)$. This field is smooth as it is parametrized by a neural network with smooth activation functions. For any empirical distribution (samples from $\rho_{t, \mu}$), we therefore obtain a vector field that is suited for inference.
>
> \
> "Parameterizing the with weight modulation as in CoLoRA [38] only applies to deterministic time-dependent dynamical systems. [...] As such, are stochastic systems omitted from the problem definition unlike the initial motivation?"
> -  We do consider stochastic systems in the numerical experiments (e.g., particles in aharmonic trap, high dimensional chaos). What we parametrize with CoLoRA is the gradient field, which is deterministic even if the system is stochastic (see line 37 in paper, as well as Section 2.1). Thus, the CoLoRA parametrization applies independent of whether the system is deterministic or stochastic. This allows a seamless treatment of deterministic and stochastic systems, which we consider a major advantage of our approach.
>
> \
> "Solving eq (7) from data is challenging and prone to potential numerical issues. [...] would be better to provide with a stability analysis on the training as mentioned in the text."
> - An empirical stability analysis is provided in Figure 2, where we show that the higher-order quadrature is essential for stabilizing the training. We stress that our results show that higher-order quadrature is not just a nice add-on to improve the accuracy a bit but that it is a core component to make the approach work in practice. In fact, when just using MC, the error blows up during training (see also the new results in the uploaded PDF).
> - A formal stability analysis is work in progress. As an outlook, we will include the following if this paper gets accepted: The difficulty is that introduction of a numerical quadrature breaks the exact correspondence between the objective
> $$
> \int_0^1 \left(\frac{1}{2}\mathbb E_{\rho_{t, \mu}}\left[|\nabla s_{t, \mu}|^2\right] + \mathbb E_{\rho_{t, \mu}} \left[ \partial_t s_{t, \mu} \right] \right) \mathrm{d} t - \mathbb E_{\rho_{t, \mu}} \left[ s_{t, \mu} \right] \big |_0^1
> $$
> and the continuity equation. Assume $\rho$ admits a density, the derivative $\partial_t$ is approximated exactly and denote by $(\dots)^n$ the value at time $t_n$ and by $w^n$ quadrature weights. The mismatch
> $$
> \left| \sum_n w^n \int s^n \partial_t \rho^n dx - \left( - \sum_n w^n \int \rho^n \partial_t s^n dx + \int \rho s \big |_0^1 dx \right) \right|
> $$
> equals the numerical integration error of $\frac{d}{dt}\int s\rho \\, dx$ in time.

---

> > ### Comment · Reviewer_zRLH · 2024-08-13
> >
> > Thank you for the rebuttal and the additional results. After reading your responses, I still have same concerns regarding the contribution of the paper.  My main issue is that while the authors claim that higher-order action matching (HOAM) is better than the baseline action matching (AM) on some parametric dynamical systems, it is yet unclear to me  how the  proposed model improves the results and help the stabilization of the training as claimed.  A clear framing of the problem setup and all the assumptions on the model would help the reader to better understand the results. I also find the implementation a bit unclear. If I understand correctly, the HOAM combines higher order numerical quadrature with MC sampling to solve (7) in the direction of time by employing a Gauss-Legendre quadrature. How can this be achieved numerically? It would be also useful to motivate the choice of the dynamical systems in the numerical experiments? Why the examples are relevant in this context is unclear. Hence, I believe all of these will require additional rewrite in multiple places. Therefore, I will keep my initial rating for now.

---

> ### Author Response · Authors · 2024-08-13
>
> > I believe all of these will require additional rewrite in multiple places.
>
> We thank the reviewer for raising these concerns. We are confident that all of the concerns are addressed in the paper, which we concisely summarize in this response. The reviewer’s comments will be helpful for guiding the final revision of this paper (if accepted) to make our points even clearer.
>
>
>
> > it is yet unclear to me how the proposed model improves the results and helps the stabilization of the training as claimed.
>
> We summarize here the reasoning for why "the proposed model improves the results and helps the stabilization of the training", which is also provided in the paper but we will make this clearer in a revision, if the paper gets accepted:
>
> 1. **Gauss-Legendre quadrature results in a more accurate estimate of the time integral when compared to Monte Carlo.** This is because it is a higher-order numerical scheme (more precisely, it allows integrating higher-degree polynomials exactly) that leads to lower quadrature errors than Monte Carlo. Besides this theoretic argument, the difference in the quadrature error can be seen empirically in our experiments in, e.g., Figure 2. (See Section 2.3.)
>
> 2. **A more accurate estimate of the time integral results in a more stable and accurate estimate of the loss.** This is supported numerically by Figure 2 (left) which shows extremely high variability in the estimates of the loss for AM and low variance estimates of loss for our proposed HOAM. This also agrees with standard results from statistical learning theory where the deviation of the empirical risk (the estimate of the loss function) from the true risk (the true loss function) directly enters in bounds of generalization errors.
>
> 3. **A more accurate estimate of the loss makes solving the optimization problem tractable.** This is supported by the numerical results in the paper, explicitly in Figure 2 (middle) and shown to be essential in global response,
>
> All this is also described in detail in the main text of our paper in Section 2.3 and numerically supported by Figure 2 and the global response. For example see line 200: "Plain uniform sampling over the data set for mini-batching can lead to poor estimates of the loss."
>
>
>
> > If I understand correctly, the HOAM combines higher order numerical quadrature with MC sampling to solve (7) in the direction of time by employing a Gauss-Legendre quadrature. How can this be achieved numerically?
>
> 1. As referenced in the main text of the paper the empirical loss is given in the Appendix in section A. This is written down as a discrete summation which can be easily implemented numerically.
> 2. For the Gauss-Legendre quadrature, the key algorithmic step is determining the roots of the Legendre polynomials, which is implemented (for example) in scipy (scipy.special.roots_legendre()). We also provide the code of our implementation, but the link is redacted for now per submission policy.
>
>
>
> > It would be also useful to motivate the choice of the dynamical systems in the numerical experiments? Why the examples are relevant in this context is unclear.
>
> 1. From a surrogate modeling perspective, the systems that we consider are exceedingly challenging because they are high dimensional, chaotic and/or stochastic. Surrogate modeling for such systems is in its infancy because the point-wise approximations of traditional surrogate modeling techniques are meaningless. We reference these attempts in detail in the introduction and literature review.
> 2. We make careful efforts to choose problem setups which are experimentally meaningful to practitioners (see for example Appendix B.2 that describes that the data are obtained from code that are used by plasma physicists). The physical relevance of our problem setups are detailed in the numerous sources we cite, see [7, 21, 34, 42, 87 Sec 2(b)(i), 58].
>
>
>
> > A clear framing of the problem setup and all the assumptions on the model would help the reader to better understand the results.
>
> 1. The problem setup is described in the Introduction section on page 1: “Given a data set of samples [...] we aim to learn a dynamical-system reduced model to rapidly predict samples that approximately follow the same law [...]”
> 2. When we make mathematical statements, we provide assumptions. For example, for the statement of ‘uniqueness’ on page 4 we assume that the density is positive and that the source term integrates to zero.
> 3. We provide an extensive appendix C-E on the connection to optimal transport and additional literature.
>
> We hope that we have been able to demonstrate that most of these concerns are addressed in the main text of the paper. If the reviewer does not have any additional concerns, we would greatly appreciate it if they would consider revising their score.

---

### Author Rebuttal · Authors · 2024-08-06

We thank the reviewers for taking the time to read the paper and very much appreciate their detailed comments. Below we provide a detailed response. We summarize here some of the main points and how we addressed them:
- We now stress that the higher-order quadrature scheme that is introduced by our approach HOAM is not just a nice add-on to improve the accuracy a bit but the core component to make the approach work in practice at all. While variants of the objective that we consider have been around in the literature for a long time, we show that higher-order quadrature in time makes the objective computationally tractable and so avoids the instabilities in training that we observe in our experiments and that agree with the instability results shown in the literature (see detailed comments below). **We further uploaded a PDF that shows more detailed results showing that HOAM is essential for stable training.**
- The problems that we examine here are intractable with current surrogate modeling and model reduction techniques (see references in paper). The improvements made with HOAM allow us to construct fast, predictive surrogate models for such problems for the first time.
- We address concerns about densely sampling in time: We consider the setting of surrogate modeling and model reduction where data are typically generated with a high-fidelity numerical model, which has to take small time steps due to numerical stability constraints. In fact, we show that HOAM performs well on a range of real problems from the physics literature (see references) for which data are generated with standard tools from the physics community. Thus, the data requirements align with what is typically available for the problems that we aim to address with our HOAM approach.

---

### Decision · Program_Chairs · 2024-09-25

**Decision:**

Accept (poster)

**Comment:**

This work is concerned with inference in dynamic systems, where the authors propose an approach that combines Monte Carlo sampling with higher-order quadrature rules. This work was reviewed by three reviewers, where one reviewer argued strongly for acceptance, one gave a borderline accept, and the third a borderline reject.

The reviewers largely agree on the strengths related to practical benefits and efficient parametric model reduction. The work shows that the model captures the temporal coupling of physical systems accurately by utilizing higher-order quadrature schemes to estimate time integrals. The idea itself can be seen as rather straightforward, but the idea is well presented.

The large spread in the reviewer scores is perhaps best explained by the presentation in the paper that slightly deviates from what one would typically expect from a NeurIPS paper. After reading the paper, I lean towards accepting this work despite the slightly mixed reviews. I trust that the authors will fix the issues (which are minor but many) that came up during the review/discussion phase.